# Registered report: A coding-independent function of gene and pseudogene mRNAs regulates tumour biology

Israr Khan[1], John Kerwin[2], Kate Owen[3], Erin Griner[3], Reproducibility Project: Cancer Biology*

[1]Alamo Laboratories Inc, San Antonio, Texas; [2]Biotechnology Research and Education Program, University of Maryland, College Park, Maryland; [3]University of Virginia, Charlottesville, Virginia

*For correspondence: tim@cos.io

Group author details
Reproducibility Project: Cancer Biology
See page 26

Competing interests:
See page 26

**Abstract** The Reproducibility Project: Cancer Biology seeks to address growing concerns about reproducibility in scientific research by conducting replications of selected experiments from a number of high-profile papers in the field of cancer biology. The papers, which were published between 2010 and 2012, were selected on the basis of citations and Altmetric scores (*Errington et al., 2014*). This Registered report describes the proposed replication plan of key experiments from 'A coding-independent function of gene and pseudogene mRNAs regulates tumour biology' by *Poliseno et al. (2010)*, published in *Nature* in 2010. The key experiments to be replicated are reported in Figures 1D, 2F-H, and 4A. In these experiments, Poliseno and colleagues report microRNAs *miR-19b* and *miR-20a* transcriptionally suppress both *PTEN* and *PTENP1* in prostate cancer cells (Figure 1D; *Poliseno et al., 2010*). Decreased expression of *PTEN* and/or *PTENP1* resulted in downregulated *PTEN* protein levels (Figure 2H), downregulation of both mRNAs (Figure 2G), and increased tumor cell proliferation (Figure 2F; *Poliseno et al., 2010*). Furthermore, overexpression of the *PTEN* 3′ UTR enhanced *PTENP1* mRNA abundance limiting tumor cell proliferation, providing additional evidence for the co-regulation of *PTEN* and *PTENP1* (Figure 4A; *Poliseno et al., 2010*). The Reproducibility Project: Cancer Biology is collaboration between the Center for Open Science and Science Exchange, and the results of the replications will be published in *eLife*.

## Introduction

The phosphatase and tensin homolog gene (*PTEN*) functions as a negative repressor of the PI3K/Akt survival pathway and is one of the most frequently deleted tumor suppressor genes in human cancer (*Stambolic et al., 1998*; *Song et al., 2012*). As a regulator of PI3K signaling, loss of *PTEN* results in over-activation of Akt, leading to unchecked cell proliferation, reduced apoptosis, and elevated tumor angiogenesis (*Stambolic et al., 1998*; *Carracedo et al., 2008*). In prostate cancer, decreases in PTEN protein expression, either by allelic deletion or functional loss caused by mutation and/or epigenetic modification, can lead to invasive prostate carcinoma (*Trotman et al., 2003*; *Phin et al., 2013*). In preclinical systems, the genetic restoration of *PTEN* induces apoptosis in cancer cell lines and has a significant negative effect on tumor growth in multiple in vivo models (*Li et al., 1998*; *Lu et al., 1999*; *Tian et al., 1999*; *Chen et al., 2011*). In contrast, clinical efforts to restore *PTEN* functionality have instead focused on targeting kinases in the PI3K pathway, including PI3K, Akt, and the mammalian target of rapamycin (*Hopkins and Parsons, 2014*). However, the development of PI3K targeting drugs has been complicated by the limited tolerability of current pharmacological treatments as well as tumor heterogeneity (*Gerlinger et al., 2012*; *Bauer et al., 2014*).

It is increasingly apparent that a complex regulatory network exists between the diverse RNA species pervasive in the human transcriptome. MicroRNAs (miRNAs) are small non-coding RNAs that bind to

complementary sequences in the 3′ untranslated regions (UTR) of target messenger RNAs (mRNA), resulting in transcriptional downregulation of the target gene (*Sen et al., 2014*). Meng and colleagues showed that *PTEN* was repressed by *miR-21*, one of the most frequently upregulated miRNAs in cancer, in hepatocarcinoma cells, suggesting that the oncogenic potential of *miR-21* occurs via the downregulation of *PTEN* expression (*Chan et al., 2005*; *Meng et al., 2006*; *Volinia et al., 2006*; *Meng et al., 2007*; *Si et al., 2007*). Several miRNAs that target *PTEN* have since been reported (*Jackson et al., 2014*; *Wang et al., 2015*). While miRNAs play a functional role in silencing target gene expression, it is proposed that miRNAs themselves are subject to regulation by competing endogenous RNA (ceRNA) species, including pseudogenes, long non-coding RNAs, and circular RNAs (*Salmena et al., 2011*; *Cesana and Daley, 2013*). In plants, for example, the non-protein coding gene *IPS1* sequesters miRNAs away from their mRNA targets, thereby leading to an accumulation of target transcripts (*Franco-Zorrilla et al., 2007*). Poliseno and colleagues proposed that pseudogenes, which are non-coding genomic DNA sequences closely related to parental genes, can modulate parental gene expression by influencing the available levels of miRNAs within a cell (*Poliseno et al., 2010*; *Cesana and Daley, 2013*). However, the extent and manner that ceRNAs can exert a consequential effect on the repression of targets for that miRNA is currently unclear (*Broderick and Zamore, 2014*). Recently, Denzler and colleagues analyzed the stoichiometric relationship of miR-122 and target sites in adult mouse liver and reported that the natural abundance of target sites exceeded miRNAs, making the ceRNA hypothesis unlikely (*Denzler et al., 2014*).

*PTENP1* is a pseudogene that shares close homology with *PTEN*, including the ability to bind miRNAs (*Fujii et al., 1999*). To determine whether *PTEN* and *PTENP1* expression levels are modulated by miRNA activity, Poliseno and colleagues first established that the *PTEN*-targeting miRNAs *miR-19b* and *miR-20a* were able to target both *PTEN* and *PTENP1* (*Poliseno et al., 2010*). As reported in Figure 1D, overexpression of *miR-19b* and *miR-20a* in prostate cancer cells resulted in a significant decrease in *PTEN* and *PTENP1* mRNA transcription. This is supported by additional studies demonstrating that overexpression of either *miR-19b* or *miR-20a* in cancer cell lines resulted in reduced *PTEN* mRNA levels and protein expression (*Luo et al., 2013*; *Tian et al., 2013*; *Wu et al., 2014*). The ability of *miR-19b* and *miR-20a* to target *PTEN* in prostate cancer was further confirmed by *Tay et al. (2011)*. These key findings established that *PTEN* and *PTENP1* are regulated by interactions with miRNA in multiple cancer cell types and will be replicated in Protocol 1.

In Figure 2F-H, Poliseno and colleagues tested the phenotypic consequences of *PTENP1* down-regulation by specifically targeting *PTEN* and/or *PTENP1* expression. Downregulation of *PTENP1* in DU145 prostate cancer cells resulted in a significant decrease in both *PTEN* and *PTENP1* mRNA levels and protein expression (Figure 2G-H; *Poliseno et al., 2010*). Furthermore, downregulation of *PTENP1* profoundly accelerated the proliferation of DU145 cells (Figure 2F), with silencing of both *PTEN* and *PTENP1* having an additive effect (*Poliseno et al., 2010*). These experiments will be replicated in Protocols 2, 3, and 4. Recently, Tay and colleagues reported that *PTEN*-ceRNAs, including *CNOT6L* and *VAPA*, phenocopied *PTENP1* activity, as downregulation of these non-coding transcripts in prostate and colon cancer cells were also able to modulate *PTEN* expression, Akt activity, and cell growth (*Tay et al., 2011*). Additionally, other *PTEN*-ceRNAs that regulate *PTEN* expression have been reported in brain, breast, and skin cancers (*Lee et al., 2010*; *Karreth et al., 2011*; *Sumazin et al., 2011*). Further to this, PTENP1 antisense RNA has been reported to regulate PTEN transcription and mRNA stability, suggesting a model where the PTENP1 pseudogene has biomodal functionality modulating PTEN (*Johnsson et al., 2013*).

As an extension of the findings reported in Figure 2 and further genomic analysis, Poliseno and colleagues demonstrated that the *PTEN* 3′ UTR regulates pseudogene expression, since over-expression of the *PTEN* 3′ UTR was found to de-repress *PTENP1* expression and inhibited DU145 proliferation (Figure 4A) (*Poliseno et al., 2010*). These experiments will be replicated in Protocols 5 and 6. These results were also confirmed by experiments by Yu and colleagues showing that overexpression of either *PTEN* or *PTENP1* suppressed renal cancer cell proliferation (*Yu et al., 2014*). Further to this, the oncosuppressive properties of overexpressing *PTENP1* 3′ UTR have been reported in various cancer cells (*Poliseno et al., 2010*; *Chen et al., 2015*; *Guo et al., 2015*).

## Materials and methods

### Protocol 1: Quantitative PCR after miR transfection

This experiment utilizes quantitative RT-PCR to analyze the effect of miR-19b or miR-20a on the mRNA levels of *PTEN* and *PTENP1*. It is a replication of Figure 1D.

## Sampling

- Experiment to be repeated a total of six times for a minimum power of 88%.
  - See 'Power calculations' section for details.
- Experiment has 5 conditions:
  - Cohort 1: siGENOME non-targeting siRNA #2 (siLUC) transfected DU145 cells.
  - Cohort 2: miR-19b transfected DU145 cells.
  - Cohort 3: miR-20a transfected DU145 cells.
  - Cohort 4: Untransfected DU145 cells (additional negative control).
  - Transfection control: siGLO RISC-free siRNA transfected DU145 cells.
- Quantitative RT-PCR performed in technical triplicate for the following genes:
  - PTEN.
  - PTENP1.
  - *ACTIN* (internal control).
  - *36B4* (additional internal control).

## Materials and reagents

| Reagent | Type | Manufacturer | Catalog # | Comments |
|---|---|---|---|---|
| DU145 cells | Cell line | ATCC | HTB-81 | – |
| RPMI 1640 medium | Cell culture | Sigma–Aldrich | R8758 | Replaces Invitrogen brand used in original study |
| Fetal bovine serum (FBS) | Cell culture | Sigma–Aldrich | F2442 | Replaces Invitrogen brand used in original study |
| L-glutamine | Cell culture | Sigma–Aldrich | G7513 | Original brand not specified |
| 100× Penicillin/streptomycin | Cell culture | Sigma–Aldrich | P4333 | Original brand not specified |
| 0.05% trypsin/0.48 mM EDTA | Cell culture | Sigma–Aldrich | T3924 | Original brand not specified |
| Phosphate buffered saline (PBS), without $MgCl_2$ and $CaCl_2$ | Cell culture | Sigma–Aldrich | D8537 | Original brand not specified |
| 12 well tissue culture dishes | Labware | Corning | 3513 | Original brand not specified |
| siGLO RISC-free siRNA | Nucleic acid | Dharmacon | D-001600-01 | – |
| siGENOME non-targeting siRNA #2 (siLUC) | Nucleic acid | Dharmacon | D-001210-02 | – |
| miRIDIAN microRNA hsa-miR-19b-3p (si-miR-19b) | Nucleic acid | Dharmacon | IH-300489-05-0002 | – |
| miRIDIAN microRNA hsa-miR-20a-5p (si-miR20a) | Nucleic acid | Dharmacon | IH-300491-05-0002 | – |
| Dharmafect 1 | Cell culture | Dharmacon | T-2001-01 | – |
| *PTENP1* forward and reverse primers | Nucleic acid | Specific brand information will be left up to the discretion of the replicating lab and recorded later | | |
| *PTEN* forward and reverse primers | Nucleic acid | | | |
| *ACTIN* forward and reverse primers | Nucleic acid | | | |
| *36B4* forward and reverse primers | Nucleic acid | | | |
| TRI reagent | Chemical | Sigma–Aldrich | T9424 | Replaces Trizol reagent from Invitrogen |
| 1-bromo-3-chloropropase | Chemical | Sigma–Aldrich | B9673 | Reagent needed from TRI reagent protocol |
| Nuclease free water | Chemical | Sigma–Aldrich | W4502 | Reagent needed from TRI reagent protocol |
| Microscope | Instrument | Zeiss | – | Original brand not specified |
| Axiovision | Software | Zeiss | – | Original brand not specified |
| DNAse I amplification grade | Chemical | Sigma–Aldrich | AMPD1 | Replaces Invitrogen brand used in original study |
| First-strand cDNA synthesis kit (includes pd(N)6 random hexamers and NotI-(dT) 18 primers) | Kit | Sigma–Aldrich | GE27-9261-01 | Replaces SuperScript II reverse transcriptase from Invitrogen used in original study |
| QuantiTect Sybr Green PCR kit | Kit | Qiagen | 204141 | – |
| Real Time System with a C1000 Thermal Cycler | Instrument | BioRad | CFX 96 | Replaces Roche Lightcycler 2.0 used in original study |

## Procedure

### Notes

- Cells will be sent for mycoplasma testing and short tandem repeat (STR) profiling.
- DU145 cells are grown in complete RPMI 1640 supplemented with 2 mM glutamine, 10% FBS, 100 U/ml penicillin and 100 μg/ml streptomycin at 37°C and 6% $CO_2$.

1. Seed $1.5 \times 10^5$ DU145 cells per well in a 12-well dish. Grow overnight.
2. Transfect with 100 nM siLuc, si-miR-19b, and si-miR-20a using 3 μl of Dharmafect 1 according to manufacturer's instructions. Transfect control cells with siGLO RISC-free control siRNA following manufacturer's instructions. Include untransfected control cells. Grow overnight.
3. Confirm that >90% of siGLO transfected control cells show fluorescence, indicating they were successfully transfected.
   a. If transfection is less than 90%, record efficiency for attempt, exclude attempt and do not continue with the rest of the procedure. Repeat procedure until >90% efficiency is obtained.
   b. If modification to transfection (step 2) is needed, record and maintain modified steps for remaining replicates.
4. 24 hr after transfection, extract total RNA from cells directly on the culture dish using TRI reagent and 1-bromo-3-chloropropane according to manufacturer's instructions.
5. Treat RNA with DNAse I following manufacturer's instructions.
   a. Record RNA concentration and purity ($A_{280}/A_{260}$).
6. Reverse transcribe 1 μg RNA/sample into cDNA using first-strand cDNA synthesis kit with primers following manufacturer's instructions.
7. Perform quantitative PCR reaction using the QuantiTect Sybr Green PCR kit:
   a. Use 2 μl of reverse transcription reaction per 20 μl real-time PCR reaction.
   b. Perform quantitative PCR for *PTEN*, *PTENP1*, *ACTIN*, and *36B4*.
      i. *PTEN* forward primer: 5′-GTTTACCGGCAGCATCAAAT-3′
      ii. *PTEN* reverse primer: 5′-CCCCCACTTTAGTGCACAGT-3′
      iii. *PTENP1* forward primer: 5′-TCAGAACATGGCATACACCAA-3′
      iv. *PTENP1* reverse primer: 5′-TGATGACGTCCGATTTTTCA-3′
      v. *ACTIN* forward primer: 5′-CATGTACGTTGCTATCCAGGC-3′
      vi. *ACTIN* reverse primer: 5′-CTCCTTAATGTCACGCACGAT-3′
      vii. *36B4* forward primer: 5′-GTGTTCGACAATGGCAGCAT-3′
      viii. *36B4* reverse primer: 5′-GACACCCTCCAGGAAGCGA-3′
         i. *36B4* primer sequences reported in *Fullwood et al. (2009)*.
   c. Do not pre-treat with uracil-N-glycosylase.
   d. All reactions should be optimized and run in technical triplicate.
8. Using *ACTIN* as an internal standard, calculate the relative *PTEN* and *PTENP1* expression for each sample using the comparative Ct method.
   a. Additionally perform normalization using *36B4* as an internal standard (additional control).
9. Repeat independently five additional times.

### Deliverables

- Data to be collected:
   ○ Images of fluorescence and phase/contrast of siGLO transfected cells.
   ○ Purity ($A_{260/280}$ ratio) and concentration of isolated total RNA from cells.
   ○ Raw data for all qPCR reactions.
   ○ Quantification of *PTEN* and *PTENP1* mRNA levels relative to *ACTIN.*
   ○ Quantification of fold change *PTEN* and *PTENP1* mRNA levels relative to siLuc transfected cells. (Compare to Figure 1D).

### Confirmatory analysis plan
This replication attempt will perform the statistical analysis listed below.

- Statistical Analysis:
   ○ Note: at the time of analysis, we will perform the Shapiro–Wilk test and generate a quantile–quantile plot to assess the normality of the data. We will also perform Levene's test to assess homoscedasticity. If the data appear skewed we will perform the appropriate transformation in

order to proceed with the proposed statistical analysis. If this is not possible we will perform the planned comparisons using the Wilcoxon–Mann Whitney test.

○ One-way MANOVA of normalized *PTEN* or *PTENP1* mRNA fold change in siLuc, 19b, or 20a siRNA transfected cells with the following planned comparisons using the Bonferroni correction:

1. *PTEN* mRNA levels of siLuc transfected cells compared to 19b transfected cells.
2. *PTEN* mRNA levels of siLuc transfected cells compared to 20a transfected cells.
3. *PTENP1* mRNA levels of siLuc transfected cells compared to 19b transfected cells.
4. *PTENP1* mRNA levels of siLuc transfected cells compared to 20a transfected cells.

■ Meta-analysis of effect sizes:

○ Compute the effect sizes of each comparison, compare them against the effect size in the original paper and use a random effects meta-analytic approach to combine the original and replication effects, which will be presented as a forest plot.

■ Additional exploratory analysis:

○ The same analysis described above will be performed with *36B4* normalized values, which serves as an independent normalization control not included in the original analysis.

## Known differences from the original study

The *PTEN* and *PTENP1* mRNA levels will be normalized with an independent control (*36B4*) in addition to *ACTIN*. All known differences are listed in the materials and reagents section above with the originally used item listed in the comments section. All differences have the same capabilities as the original and are not expected to alter the experimental design.

## Provisions for quality control

The cell line used in this experiment will undergo STR profiling to confirm its identity and will be sent for mycoplasma testing to ensure there is no contamination. Transfection efficiency will be recorded for each replicate and any transfection that does not contain >90% efficiency will be excluded and not continue through the rest of the procedure. If the efficiency in the first attempt(s) does not obtain >90%, then any modifications to the transfection protocol will be recorded and the procedure will be maintained for the remaining replicates. The sample purity ($A_{260/280}$ ratio) of the isolated RNA from each sample will be reported. The *PTEN* and *PTENP1* mRNA levels will be normalized with an independent control (*36B4*). All the raw data, including the analysis files, will be uploaded to the project page on the Open Science Framework (OSF) (https://osf.io/yyqas) and made publically available.

## Protocol 2: Cell growth assay following siRNA transfection

This experiment tests the effect of siRNA mediated depletion of *PTEN*, *PTENP1*, or both on the growth of DU145 cells. It is a replication of Figure 2F.

### Sampling

■ Experiment to be repeated a total of five times for a minimum power of 94%.
○ See 'Power calculations' section for details.

■ Experiment has 6 conditions:
○ Cohort 1: siGENOME non-targeting siRNA #2 (siLUC) transfected DU145 cells.
○ Cohort 2: siPTEN Smartpool (targets *PTEN* and *PTENP1*) transfected DU145 cells.
○ Cohort 3: siPTEN transfected DU145 cells.
○ Cohort 4: siPTENP1 transfected DU145 cells.
○ Cohort 5: Untransfected DU145 cells (additional negative control).
○ Transfection control: siGLO RISC-free siRNA transfected DU145 cells.

■ Each cohort is harvested on the following days performed in technical triplicate:
○ Day 0 (after O/N incubation).
○ Day 1.
○ Day 2.
○ Day 3.
○ Day 4.
○ Day 5.

# Materials and reagents

| Reagent | Type | Manufacturer | Catalog # | Comments |
|---|---|---|---|---|
| DU145 cells | Cell line | ATCC | HTB-81 | – |
| RPMI 1640 medium | Cell culture | Sigma–Aldrich | R8758 | Replaces Invitrogen brand used in original study |
| Fetal bovine serum (FBS) | Cell culture | Sigma–Aldrich | F2442 | Replaces Invitrogen brand used in original study |
| L-glutamine | Cell culture | Sigma–Aldrich | G7513 | Original brand not specified |
| 100× Penicillin/streptomycin | Cell culture | Sigma–Aldrich | P4333 | Original brand not specified |
| 0.05% trypsin/0.48 mM EDTA | Cell culture | Sigma–Aldrich | T3924 | Original brand not specified |
| Phosphate buffered saline (PBS), without $MgCl_2$ and $CaCl_2$ | Cell culture | Sigma–Aldrich | D8537 | Original brand not specified |
| 12 well tissue culture dishes | Labware | Corning | 3513 | Original brand not specified |
| siGLO RISC-free siRNA | Nucleic acid | Dharmacon | D-001600-01 | – |
| siGENOME non-targeting siRNA #2 (siLUC) | Nucleic acid | Dharmacon | D-001210-02 | – |
| siPTEN | Nucleic acid | Dharmacon | Custom | See Supplemental Figure 6 of original paper for sequence |
| ON-TARGETplus siPTEN Smartpool | Nucleic acid | Dharmacon | L-003023-00 | Composed of: J-003023-09; J-003023-10; J-003023-11; J-003023-12 |
| siPTENP1 | Nucleic acid | Dharmacon | Custom | See Supplemental Figure 6 of original paper for sequence |
| Dharmafect 1 | Cell culture | Dharmacon | T-2001-01 | – |
| Microscope | Instrument | Olympus | LX81 | Original brand not specified |
| Crystal violet | Dye | Sigma–Aldrich | C0775 | Original brand not specified |
| Formalin | Chemical | Specific brand information will be left up to the discretion of the replicating lab and recorded later | | |
| Acetic acid | Chemical | | | |
| Methanol | Chemical | | | |
| Spectrophotometer capable of reading at 590 nm (or 595 nm) | Instrument | BioTek Instruments | Synergy 2 (SLFA configuration) | Original brand not specified |

## Procedure

### Note

- All cells will be sent for mycoplasma testing and STR profiling.
- DU145 cells grown in complete RPMI 1640: RPMI 1640 supplemented with 2 mM glutamine, 10% FBS, 100 U/ml penicillin and 100 µg/ml streptomycin at 37°C and 6% $CO_2$.

1. Seed $1.5 \times 10^5$ DU145 cells per well in a 12-well dish. Grow overnight.
2. Transfect with 100 nM siRNAs (siPTEN, siPTENP1, siPTEN Smartpool (siPTEN and PTENP1), or siLuc in separate wells) using Dharmafect 1 according to manufacturer's instructions or leave untransfected. Transfect control cells with siGLO RISC-free control siRNA according to manufacturer's instructions. Grow overnight.
3. Confirm that >90% of siGLO transfected control cells show fluorescence, indicating they were successfully transfected.
   a. If transfection is less than 90%, record efficiency for attempt, exclude attempt and do not continue with the rest of the procedure. Repeat procedure until >90% efficiency is obtained.
   b. If modification to transfection (step 2) is needed, record and maintain modified steps for remaining replicates.
4. The day after transfection, resuspend $2 \times 10^5$ siLuc, siPTEN, siPTENP1, siPTEN/PTENP1, or untransfected cells in 50 ml fresh media. Seed three wells of six sets of 12-well plates with 2 ml of each cell line. Each set of 12 well plates should have three wells of each cell line. Incubate overnight.
5. Fix one plate every 24 hr starting after overnight incubation (the first plate fixed will be called day 0).
   a. Wash wells once in PBS.

 b. Fix wells with 10% formalin for 10 min at room temperature.
 c. Store plates in PBS at 4°C.
 d. All wells should be fixed by day 6.
6. Stain cells with 0.1% crystal violet, 20% methanol for 15 min. Wash cells.
7. Lyse all wells with 10% acetic acid for 10 min.
8. Read optical density at 590 or 595 nm.
 a. Reading can be done at 595 nm if 590 is not available.
9. Repeat independently four additional times.

## Deliverables

- Data to be collected:
  ○ Images of fluorescence and phase/contrast of siGLO transfected cells.
  ○ Raw data of absorbance from plate reader.
  ○ Graph of relative cell number for each cell line over time. (Compare to Figure 2F).

## Confirmatory analysis plan
This replication attempt will perform the following statistical analysis listed below.

- Statistical Analysis:
  ○ Note: at the time of analysis, we will perform the Shapiro–Wilk test and generate a quantile–quantile plot to assess the normality of the data. We will also perform Levene's test to assess homoscedasticity. If the data appear skewed we will perform the appropriate transformation in order to proceed with the proposed statistical analysis. If this is not possible we will perform the equivalent non-parametric test.
  ○ Two-way ANOVA comparing Day 5 absorbance in siLuc, siPTEN, siPTENP1, or siPTEN/PTENP1 transfected cells with the following planned comparisons using the Bonferroni correction:
  1. siLuc compared to siPTEN.
  2. siLuc compared to siPTENP1.
  3. siLuc compared to siPTEN/PTENP1.
  4. siPTEN/PTENP1 compared to siPTEN.
  5. siPTEN/PTENP1 compared to siPTENP1.
  ○ Two-way ANOVA comparing area under the curve (AUC) measurements (determined from day 0, 1, 2, 3, 4, and 5 for each replicate) from absorbance in siLuc, siPTEN, siPTENP1, or siPTEN/PTENP1 transfected cells with the following planned comparisons using the Bonferroni correction.
  1. siLuc compared to siPTEN.
  2. siLuc compared to siPTENP1.
  3. siLuc compared to siPTEN/PTENP1.
  4. siPTEN/PTENP1 compared to siPTEN.
  5. siPTEN/PTENP1 compared to siPTENP1.
- Meta-analysis of effect sizes:
  ○ Compute the effect sizes of each comparison, compare them against the effect size in the original paper and use a random effects meta-analytic approach to combine the original and replication effects, which will be presented as a forest plot.

## Known differences from the original study
All known differences are listed in the materials and reagents section above with the originally used item listed in the comments section. All differences have the same capabilities as the original and are not expected to alter the experimental design.

## Provisions for quality control
The cell line used in this experiment will undergo STR profiling to confirm its identity and will be sent for mycoplasma testing to ensure there is no contamination. Transfection efficiency will be recorded for each replicate and any transfection that does not contain >90% efficiency will be excluded and not continue through the rest of the procedure. If the efficiency in the first attempt(s) does not obtain >90%, then any modifications to the transfection protocol will be recorded and the procedure will be maintained for the remaining replicates. All the raw data, including the analysis files, will be uploaded to the project page on the OSF (https://osf.io/yyqas) and made publically available.

## Protocol 3: Quantitative PCR following transfected with siRNA against PTEN and/or PTENP1

This experiment analyzes the effect of depletion of *PTEN*, *PTENP1*, or both on the mRNA expression of *PTEN* or *PTENP1*. Quantitative real time PCR is utilized to assess the levels of expression following transfection of siRNA. This protocol is a replication of Figure 2G.

### Sampling

- Experiment to be repeated a total of five times for a minimum power of 89%.
  - See 'Power calculations' section for details.
- Experiment has 6 conditons:
  - Cohort 1: Uninfected DU145 cells (additional negative control).
  - Cohort 1: siGENOME non-targeting siRNA #2 (siLUC) transfected DU145 cells.
  - Cohort 2: siPTEN Smartpool (targets *PTEN* and *PTENP1*) transfected DU145 cells.
  - Cohort 3: siPTEN transfected DU145 cells.
  - Cohort 4: siPTENP1 transfected DU145 cells.
  - Cohort 5: Uninfected DU145 cells (additional negative control).
  - Transfection control: siGLO RISC-free siRNA transfected DU145 cells.
- Quantitative RT-PCR performed in technical triplicate for the following genes:
  - *PTEN.*
  - PTEN1P.
  - *ACTIN* (internal control).
  - *36B4* (additional internal control).

### Materials and reagents

| Reagent | Type | Manufacturer | Catalog # | Comments |
|---------|------|-------------|-----------|----------|
| DU145 cells | Cell line | ATCC | HTB-81 | – |
| RPMI 1640 medium | Cell culture | Sigma–Aldrich | R8758 | Replaces Invitrogen brand used in original study |
| Fetal bovine serum (FBS) | Cell culture | Sigma–Aldrich | F2442 | Replaces Invitrogen brand used in original study |
| L-glutamine | Cell culture | Sigma–Aldrich | G7513 | Original brand not specified |
| 100× Penicillin/streptomycin | Cell culture | Sigma–Aldrich | P4333 | Original brand not specified |
| 0.05% trypsin/0.48 mM EDTA | Cell culture | Sigma–Aldrich | T3924 | Original brand not specified |
| Phosphate buffered saline (PBS), without MgCl$_2$ and CaCl$_2$ | Cell culture | Sigma–Aldrich | D8537 | Original brand not specified |
| 12 well tissue culture dishes | Labware | Corning | 3513 | Original brand not specified |
| siGLO RISC-free siRNA | Nucleic acid | Dharmacon | D-001600-01 | – |
| siGENOME non-targeting siRNA #2 (siLUC) | Nucleic acid | Dharmacon | D-001210-02 | – |
| ON-TARGETplus siPTEN Smartpool | Nucleic acid | Dharmacon | L-003023-00 | Composed of: J-003023-09; J-003023-10; J-003023-11; J-003023-12 |
| siPTEN | Nucleic acid | Dharmacon | Custom | See Supplemental Figure 6 of original paper for sequence |
| siPTENP1 | Nucleic acid | Dharmacon | Custom | See Supplemental Figure 6 of original paper for sequence |
| Dharmafect 1 | Cell culture | Dharmacon | T-2001-01 | – |
| Microscope | Instrument | Olympus | LX81 | Original brand not specified |
| *ACTIN* forward and reverse primers | Nucleic acid | Specific brand information will be left up to the discretion of the replicating lab and recorded later | | |
| *PTEN* forward and reverse primers | Nucleic acid | | | |
| *PTENP1* forward and reverse primers | Nucleic acid | | | |
| *36B4* forward and reverse primers | Nucleic acid | | | |
| TRI reagent | Chemical | Sigma–Aldrich | T9424 | Replaces Trizol reagent from Invitrogen |
| 1-bromo-3-chloropropase | Chemical | Sigma–Aldrich | B9673 | Reagent needed from TRI reagent protocol |

*Continued on next page*

*Continued*

| Reagent | Type | Manufacturer | Catalog # | Comments |
|---|---|---|---|---|
| Nuclease free water | Chemical | Sigma–Aldrich | W4502 | Reagent needed from TRI reagent protocol |
| DNase I amplification grade | Chemical | Sigma–Aldrich | AMPD1 | Replaces Invitrogen brand used in original study |
| First-strand cDNA synthesis kit (includes pd(N)6 random hexamers and NotI-(dT) 18 primers) | Kit | Sigma–Aldrich | GE27-9261-01 | Replaces SuperScript II reverse transcriptase from Invitrogen used in original study |
| QuantiTect Sybr Green PCR kit | Kit | Qiagen | 204141 | – |
| Real-time PCR machine | Instrument | Applied Biosystems | 7500 | Replaces Lightcycler 2.0 from Roche used in original study |

## Procedure

### Note

- All cells will be sent for mycoplasma testing and STR profiling.
- DU145 cells grown in complete RPMI 1640: RPMI 1640 supplemented with 2 mM glutamine, 10% FBS, 100 U/ml penicillin and 100 μg/ml streptomycin at 37°C and 6% $CO_2$.

1. Seed $1.5 \times 10^5$ DU145 cells per well in a 12-well dish. Grow overnight.
2. Transfect with 100 nM siRNAs (siPTEN, siPTENP1, siPTEN Smartpool (siPTEN/PTENP1), or siLuc in separate wells) using Dharmafect 1 according to manufacturer's instructions or leave untransfected. Transfect control cells with siGLO RISC-free control siRNA according to manufacturer's instructions. Grow overnight.
3. Confirm that >90% of siGLO transfected control cells show fluorescence, indicating they were successfully transfected.
   a. If transfection is less than 90%, record efficiency for attempt, exclude attempt and do not continue with the rest of the procedure. Repeat procedure until >90% efficiency is obtained.
   b. If modification to transfection (step 2) is needed, record and maintain modified steps for remaining replicates.
4. 24 hr after transfection, extract total RNA directly on the culture dish using TRI reagent and 1-bromo-3-chloropropane according to manufacturer's instructions.
5. Treat RNA with DNAse following manufacturer's instructions.
6. Reverse transcribe 1 μg RNA/sample into cDNA using first-strand cDNA synthesis kit with primers following manufacturer's instructions.
   a. Record RNA concentration and purity ($A_{280}/A_{260}$).
7. Perform quantitative PCR reaction using the QuantiTect Sybr Green PCR kit:
   a. Use 2 μl of reverse transcription reaction per 20 μl real-time PCR reaction.
   b. Perform quantitative PCR for *PTEN*, *PTENP1*, *ACTIN*, and *36B4.*
      i. *PTEN* forward primer: 5′-GTTTACCGGCAGCATCAAAT-3′
      ii. *PTEN* reverse primer: 5′-CCCCCACTTTAGTGCACAGT-3′
      iii. *PTENP1* forward primer: 5′-TCAGAACATGGCATACACCAA-3′
      iv. *PTENP1* reverse primer: 5′-TGATGACGTCCGATTTTTCA-3′
      v. *ACTIN* forward primer: 5′-CATGTACGTTGCTATCCAGGC-3′
      vi. *ACTIN* reverse primer: 5′-CTCCTTAATGTCACGCACGAT-3′
      vii. *36B4* forward primer: 5′-GTGTTCGACAATGGCAGCAT-3′
      viii. *36B4* reverse primer: 5′-GACACCCTCCAGGAAGCGA-3′
         i. *36B4* primer sequences reported in *Fullwood et al. (2009)*.
   c. Do not pre-treat with uracil-N-glycosylase.
   d. All reactions should be optimized and run in technical triplicate.
8. Using *ACTIN* as an internal standard, calculate the relative *PTEN* and *PTENP1* expression for each sample using the comparative Ct method.
   a. Additionally perform normalization using *36B4* as an internal standard (additional control).
9. Repeat independently four additional times.

### Deliverables

- Data to be collected:
  ○ Images of fluorescence and phase/contrast of siGLO transfected cells.

○ Purity ($A_{260/280}$ ratio) and concentration of isolated total RNA from cells.
○ Raw data for all qPCR reactions.
○ Quantification of *PTEN* and *PTENP1* mRNA levels relative to *ACTIN* or *36B4*.
○ Quantification of fold change *PTEN* and *PTENP1* mRNA levels relative to siLuc transfected cells.
○ Graph of fold change *PTEN* mRNA expression relative to siLuc. (Compare to Figure 2G, left).
○ Graph of fold change *PTENP1* mRNA expression relative to siLuc. (Compare to Figure 2G, right).

## Confirmatory analysis plan

This replication attempt will perform the following statistical analysis listed below.

- Statistical Analysis:
  ○ Note: at the time of analysis, we will perform the Shapiro–Wilk test and generate a quantile–quantile plot to assess the normality of the data. We will also perform Levene's test to assess homoscedasticity. If the data appear skewed we will perform the appropriate transformation in order to proceed with the proposed statistical analysis. If this is not possible we will perform the planned comparisons using the Wilcoxon–Mann Whitney test.
  ○ One-way MANOVA of *PTEN* and *PTENP1* mRNA levels in siLuc, siPTEN, siPTENP1, or siPTEN/PTENP1 siRNA transfected cells with the following planned comparisons using the Bonferroni correction:
    1. *PTEN* mRNA levels of siLuc transfected cells compared to siPTEN transfected cells.
    2. *PTEN* mRNA levels of siLuc transfected cells compared to siPTENP1 transfected cells.
    3. *PTEN* mRNA levels of siLuc transfected cells compared to siPTEN/PTENP1 transfected cells.
    4. *PTENP1* mRNA levels of siLuc transfected cells compared to siPTEN transfected cells.
    5. *PTENP1* mRNA levels of siLuc transfected cells compared to siPTENP1 transfected cells.
    6. *PTENP1* mRNA levels of siLuc transfected cells compared to siPTEN/PTENP1 transfected cells.
- Meta-analysis of effect sizes:
  ○ Compute the effect sizes of each comparison, compare them against the effect size in the original paper and use a random effects meta-analytic approach to combine the original and replication effects, which will be presented as a forest plot.
- Additional exploratory analysis:
  ○ The same analysis described above will be performed with *36B4* normalized values, which serves as an independent normalization control not included in the original analysis.

## Known differences from the original study

The *PTEN* and *PTENP1* mRNA levels will be normalized with an independent control (*36B4*) in addition to *ACTIN*. All known differences are listed in the materials and reagents section above with the originally used item listed in the comments section. All differences have the same capabilities as the original and are not expected to alter the experimental design.

## Provisions for quality control

The cell line used in this experiment will undergo STR profiling to confirm their identity and will be sent for mycoplasma testing to ensure there is no contamination. Transfection efficiency will be recorded for each replicate and any transfection that does not contain >90% efficiency will be excluded and not continue through the rest of the procedure. If the efficiency in the first attempt(s) does not obtain >90%, then any modifications to the transfection protocol will be recorded and the procedure will be maintained for the remaining replicates. The sample purity ($A_{260/280}$ ratio) of the isolated RNA from each sample will be reported. The *PTEN* and *PTENP1* mRNA levels will be normalized with an independent control (*36B4*). All the raw data, including the analysis files, will be uploaded to the project page on the OSF (https://osf.io/yyqas) and made publically available.

## Protocol 4: Western blot of cells transfected with siRNA

This experiment utilizes western blot to assess the protein levels of PTEN after depletion of *PTEN*, *PTENP1*, or both. It is a replication of Figure 2H.

## Sampling

- Experiment to be repeated a total of five times for a minimum power of 80%. The original data are qualitative, thus to determine an appropriate number of replicates to initially perform, sample sizes based on a range of potential variance was determined.

○ See 'Power calculations' section for details.
■ Experiment has 6 conditons:
  ○ Cohort 1: siGENOME non-targeting siRNA #2 (siLUC) transfected DU145 cells.
  ○ Cohort 2: siPTEN Smartpool (targets *PTEN* and *PTENP1*) transfected DU145 cells.
  ○ Cohort 3: siPTEN transfected DU145 cells.
  ○ Cohort 4: siPTENP1 transfected DU145 cells.
  ○ Cohort 5: Uninfected DU145 cells (additional negative control).
  ○ Transfection control: siGLO RISC-free siRNA transfected DU145 cells.
■ Western blots performed for:
  ○ PTEN.
  ○ Hsp90 (loading control).

## Materials and reagents

| Reagent | Type | Manufacturer | Catalog # | Comments |
|---|---|---|---|---|
| DU145 cells | Cell line | ATCC | HTB-81 | – |
| RPMI 1640 medium | Cell culture | Sigma–Aldrich | R8758 | Replaces Invitrogen brand used in original study |
| Fetal bovine serum (FBS) | Cell culture | Sigma–Aldrich | F2442 | Replaces Invitrogen brand used in original study |
| L-glutamine | Cell culture | Sigma–Aldrich | G7513 | Original brand not specified |
| 100× Penicillin/streptomycin | Cell culture | Sigma–Aldrich | P4333 | Original brand not specified |
| 0.05% trypsin/0.48 mM EDTA | Cell culture | Sigma–Aldrich | T3924 | Original brand not specified |
| Phosphate buffered saline (PBS), without $MgCl_2$ and $CaCl_2$ | Cell culture | Sigma–Aldrich | D8537 | Original brand not specified |
| 6 well tissue culture dishes | Labware | Corning | 3516 | Original brand not specified |
| siGLO RISC-free siRNA | Nucleic acid | Dharmacon | D-001600-01 | – |
| siGENOME non-targeting siRNA #2 (siLUC) | Nucleic acid | Dharmacon | D-001210-02 | – |
| ON-TARGETplus siPTEN Smartpool | Nucleic acid | Dharmacon | L-003023-00 | Composed of: J-003023-09; J-003023-10; J-003023-11; J-003023-12 |
| siPTEN | Nucleic acid | Dharmacon | Custom | See Supplemental Figure 6 of original paper for sequence |
| siPTENP1 | Nucleic acid | Dharmacon | Custom | See Supplemental Figure 6 of original paper for sequence |
| Dharmafect 1 | Cell culture | Dharmacon | T-2001-01 | – |
| Microscope | Instrument | Olympus | LX81 | Original brand not specified |
| Rabbit anti-PTEN (clone 138G6) monoclonal antibody | Antibodies | Cell Signaling | 9559 | – |
| Mouse anti-Hsp90 (clone 68) antibody | Antibodies | Becton Dickinson | 610419 | Original catalog number not specified |
| Secondary antibody (anti-rabbit IgG) | Antibodies | Cell Signaling | 7074 | Original brand not specified |
| Secondary antibody (anti-mouse IgG) | Antibodies | Cell Signaling | 7076 | Original brand not specified |
| ECL DualVue Western Markers (15–150 kDa) | Western blot reagent | Sigma–Aldrich | GERPN810 | Original brand not specified |
| Tris | Chemical | | | Specific brand information will be left up to the discretion of the replicating lab and recorded later |
| EDTA | Chemical | | | |
| $MgCl_2$ | Chemical | | | |
| NaCl | Chemical | | | |
| $NP_40$ | Chemical | | | |
| β-glycerophsphate | Chemical | | | |
| $NaVO_4$ | Chemical | | | |
| NaF | Chemical | | | |

*Continued on next page*

*Continued*

| Reagent | Type | Manufacturer | Catalog # | Comments |
|---|---|---|---|---|
| Protease inhibitor cocktail (mammalian) | Inhibitor | Sigma–Aldrich | P8340 | Original brand not specified |
| Sonifier | Instrument | Branson Digital | n/a | Original brand not specified |
| Bradford Reagent | Reporter assay | Sigma–Aldrich | B6916 | Original brand not specified |
| TruPAGE LDS sample buffer (4×) | Buffer | Sigma–Aldrich | PCG3009 | Original brand not specified |
| TruPAGE DTT sample reducer (10×) | Buffer | Sigma–Aldrich | PCG3005 | – |
| XCell SureLOCK Mini-cell system | Instrument | Life Technologies | n/a | Original brand not specified |
| 4–12% TruPAGE SDS-PAGE gel | Western blot reagent | Sigma–Aldrich | PCG2003 | Replaces NuPage gels |
| TruPAGE TEA-Tricine SDS running buffer (20×) | Buffer | Sigma–Aldrich | PCG3001 | Original brand not specified |
| Xcell II Blot Module | Instrument | Life Technologies | n/a | Original brand not specified |
| Hybond ECL nitrocellulose membrane | Western blot reagent | Sigma–Aldrich | GERPN2020D | Original brand not specified |
| TruPAGE transfer buffer (20×) | Buffer | Sigma–Aldrich | PCG3011 | Original brand not specified |
| Ponceau S solution | Buffer | Sigma–Aldrich | P7170 | – |
| 10× Tris buffered saline (TBS) | Buffer | Sigma–Aldrich | T5912 | Original brand not specified |
| ECL Prime Western blotting system | Detection assay | Sigma–Aldrich | GERPN2232 | Original brand not specified |
| Image J | Software | NIH | Version 10.2 | – |

## Procedure

### Note

- All cells will be sent for mycoplasma testing and STR profiling.
- DU145 cells grown in complete RPMI 1640: RPMI 1640 supplemented with 2 mM glutamine, 10% FBS, 100 U/ml penicillin and 100 µg/ml streptomycin at 37°C and 6% $CO_2$.

1. Seed $3.75 \times 10^5$ DU145 cells per well in a 6-well dish. Grow overnight.
2. Transfect with 100 nM siRNAs (siPTEN, siPTENP1, siPTEN Smartpool (siPTEN/PTENP1), or siLuc in separate wells) using Dharmafect 1 according to manufacturer's instructions or leave untransfected. Transfect control cells with siGLO RISC-free control siRNA according to manufacturer's instructions. Grow overnight.
3. Confirm that >90% of siGLO transfected control cells show fluorescence, indicating they were successfully transfected.
   a. If transfection is less than 90%, record efficiency for attempt, exclude attempt and do not continue with the rest of the procedure. Repeat procedure until >90% efficiency is obtained.
   b. If modification to transfection (step 2) is needed, record and maintain modified steps for remaining replicates.
4. 48 hr after transfection lyse cells transfected with siRNAs and uninfected cells in lysis buffer on ice for 30 min.
   a. Lysis buffer: 50 mM Tris pH8.0, 1 mM EDTA, 1 mM $MgCl_2$, 150 mM NaCl, 1% NP-40, 1 mM β-glycerophosphate, 1 mM $Na_3VO_4$, 1 mM NaF, protease inhibitors.
5. Gently sonicate protein lysate for 3 to 4 bursts for 5 to 10 s. Clear lysate by centrifugation at 10,000×*g* for 10 min at 4°C.
6. Perform Bradford protein determination assay following manufacturer's instructions.
7. Separate 30 µg of protein (in 1× sample buffer and sample reducer) per lane on a 4–12% Tris Glycine SDS-PAGE gel with protein ladder following manufacturer's instructions.
   a. Sample run per gel:
      i. Protein molecular weight marker.
      ii. Untransfected DU145 cells.
      iii. DU145 cells transfected with siGENOME non-targeting siRNA #2.
      iv. DU145 cells transfected with siPTEN.
      v. DU145 cells transfected with siPTENP1.
      vi. DU145 cells transfected with siPTEN/PTENP1.

8. Transfer to nitrocellulose membrane (pre-wetted with methanol before use) at 25 V constant for 1–2 hr in 1× transfer buffer with 20% methanol following manufacturer's instructions.
   a. After transfer, stain membrane with Ponceau S solution following manufacturer's instructions to visualize transferred protein. Image membrane, then wash out the Ponceau stain (additional quality control step).
9. Perform western blotting with the following antibodies following manufacturer's instructions. Use 1× TBS for washes and blocking reagent recommended by manufacturer.
   a. rabbit anti-PTEN; use at 1:1000 dilution; 54 kDa.
   b. mouse anti-Hsp90; use at 1:1000 dilution; 90 kDa.
10. Detect signal with appropriate HRP conjugated secondary antibody followed by chemiluminescence following manufacturer's instructions.
11. Analyze scanned images using Image J software.
    a. Equal-sized regions of interest (ROI) will be positioned on specific bands.
    b. Background will be located within each individual lane but not occupied by any other discrete band.
    c. Subtract background pixel intensity from ROI pixel intensity.
    d. Normalize PTEN values by Hsp90 values from the same sample.
12. Repeat independently four additional times.

## Deliverables

- Data to be collected:
  ○ Images of fluorescence and phase/contrast of siGLO transfected cells.
  ○ Images of Ponceau stained membranes and full films for all western blots with ladder. (Compare to Figure 2H).
  ○ Raw data file of ROI and background pixel intensities.
  ○ Normalize PTEN values for each sample.

## Confirmatory analysis plan

This replication attempt will perform the following statistical analysis listed below.

- Statistical Analysis:
  ○ Note: at the time of analysis, we will perform the Shapiro–Wilk test and generate a quantile–quantile plot to assess the normality of the data. We will also perform Levene's test to assess homoscedasticity. If the data appear skewed we will perform the appropriate transformation in order to proceed with the proposed statistical analysis. If this is not possible we will perform the equivalent non-parametric test.
  ○ Two-way ANOVA of normalized PTEN levels in siLuc, siPTEN, siPTENP1, or siPTEN/PTENP1 siRNA transfected cells with the following planned comparisons using the Bonferroni correction:
  1. siLuc compared to siPTEN.
  2. siLuc compared to siPTENP1.
  3. siLuc compared to siPTEN/PTENP1.
  4. siPTEN/PTENP1 compared to siPTEN.
  5. siPTEN/PTENP1 compared to siPTENP1.
- Meta-analysis of effect sizes:
  ○ The replication data (mean and 95% confidence interval) will be plotted with the original reported data value plotted as a single point on the same plot for comparison.

### Known differences from the original study

The original study used 12 well plates seeded with $1.5 \times 10^5$ DU145 cells per well, which was increased 2.5× to account for the difference in cell surface area. All known differences are listed in the materials and reagents section above with the originally used item listed in the comments section. All differences have the same capabilities as the original and are not expected to alter the experimental design.

### Provisions for quality control

The cell line used in this experiment will undergo STR profiling to confirm their identity and will be sent for mycoplasma testing to ensure there is no contamination. Transfection efficiency will be recorded for each replicate and any transfection that does not contain >90% efficiency will be excluded and not continue through the rest of the procedure. If the efficiency in the first attempt

(s) does not obtain >90%, then any modifications to the transfection protocol will be recorded and the procedure will be maintained for the remaining replicates. Ponceau stained membranes will be used to assess completeness of transfer. All the raw data, including the analysis files, will be uploaded to the project page on the OSF (https://osf.io/yyqas) and made publically available.

## Protocol 5: Quantitative PCR following PTEN 3′ UTR transfection

This experiment tests the effect of expressing the 3′ UTR of *PTENP1* on mRNA expression levels of *PTENP1*. It is a replication of the left panel of Figure 4A.

### Sampling

- Experiment to be repeated a total of three times for a minimum power of 98%.
  - See 'Power calculations' section for details.
- Experiment has 3 conditions:
  - Cohort 1: pCMV transfected DU145 cells.
  - Cohort 2: pCMV/*PTEN* 3′ UTR transfected DU145 cells.
  - Cohort 3: Uninfected DU145 cells (additional negative control).
- Quantitative RT-PCR performed in technical triplicate for the following genes:
  - *PTENP1*.
  - *ACTIN* (internal control).
  - *36B4* (additional internal control).

### Materials and reagents

| Reagent | Type | Manufacturer | Catalog # | Comments |
|---|---|---|---|---|
| DU145 cells | Cell line | ATCC | HTB-81 | – |
| RPMI 1640 medium | Cell culture | Sigma–Aldrich | R8758 | Replaces Invitrogen brand used in original study |
| Fetal bovine serum (FBS) | Cell culture | Sigma–Aldrich | F2442 | Replaces Invitrogen brand used in original study |
| L-glutamine | Cell culture | Sigma–Aldrich | G7513 | Original brand not specified |
| 100× Penicillin/streptomycin | Cell culture | Sigma–Aldrich | P4333 | Original brand not specified |
| 0.05% trypsin/0.48 mM EDTA | Cell culture | Sigma–Aldrich | T3924 | Original brand not specified |
| Phosphate buffered saline (PBS), without $MgCl_2$ and $CaCl_2$ | Cell culture | Sigma–Aldrich | D8537 | Original brand not specified |
| 60 mm tissue culture dishes | Labware | Corning | 430166 | Original brand not specified |
| Endo-free maxiprep kit | Kit | Sigma–Aldrich | NA0400 | – |
| pCMV (empty vector) | DNA construct | Original lab | n/a | From original lab |
| pCMV/*PTEN* 3′ UTR | DNA construct | Original lab | n/a | From original lab |
| Effectene | Cell culture | Qiagen | 301425 | Original brand not specified |
| *PTENP1* forward and reverse primers | Nucleic acid | Specific brand information will be left up to the discretion of the replicating lab and recorded later | | |
| *ACTIN* forward and reverse primers | Nucleic acid | | | |
| *36B4* forward and reverse primers | Nucleic acid | | | |
| TRI reagent | Chemical | Sigma–Aldrich | T9424 | Replaces Trizol reagent from Invitrogen |
| 1-bromo-3-chloropropase | Chemical | Sigma–Aldrich | B9673 | Reagent needed from TRI reagent protocol |
| Nuclease free water | Chemical | Sigma–Aldrich | W4502 | Reagent needed from TRI reagent protocol |
| DNAse I amplification grade | Chemical | Sigma–Aldrich | AMPD1 | Replaces Invitrogen brand used in original study |
| First-strand cDNA synthesis kit (includes pd(N)6 random hexamers and NotI-(dT) 18 primers) | Kit | Sigma–Aldrich | GE27-9261-01 | Replaces SuperScript II reverse transcriptase from Invitrogen used in original study |
| QuantiTect Sybr Green PCR kit | Kit | Qiagen | 204141 | – |
| Real-time PCR system | Instrument | Applied Biosystems | 7500 Fast | Replaces Roche Lightcycler 2.0 used in original study |

## Procedure

### Note

- All cells will be sent for mycoplasma testing and STR profiling.
- DU145 cells grown in complete RPMI 1640: RPMI 1640 supplemented with 2 mM glutamine, 10% FBS, 100 U/ml penicillin and 100 μg/ml streptomycin at 37°C and 6% $CO_2$.

1. Grow and prepare endotoxin-free plasmid constructs following manufacturer's instructions for an endotoxin-free plasmid maxiprep kit.
   a. pCMV (empty vector).
   b. pCMV/*PTEN* 3′ UTR.
      i. Sequence gene of interest in each plasmid and run whole plasmids on agarose gel to confirm vector integrity.
2. Seed $3.5 \times 10^5$ DU145 cells per dish in 6 cm dishes. Grow overnight.
3. Transfect with pCMV or pCMV/*PTEN* 3′ UTR plasmids using Effectene according to manufacturer's instructions and recommended DNA and reagent amounts.
4. 24 hr after transfection, extract total RNA from each cohort directly on the culture dish using TRI reagent and 1-bromo-3-chloropropane according to manufacturer's instructions.
5. Treat RNA with DNase I following manufacturer's instructions.
6. Reverse transcribe 1 μg RNA/sample into cDNA using first-strand cDNA synthesis kit with primers following manufacturer's instructions.
   a. Record RNA concentration and purity ($A_{280}/A_{260}$).
7. Perform quantitative PCR reaction using the QuantiTect SYBR Green PCR kit:
   a. Use 2 μl of reverse transcription reaction per 20 μl real-time PCR reaction.
   b. Perform quantitative PCR for *PTENP1*, *ACTIN*, and *36B4*.
      i. *PTENP1* forward primer: 5′-TCAGAACATGGCATACACCAA-3′
      ii. *PTENP1* reverse primer: 5′-TGATGACGTCCGATTTTTCA-3′
      iii. *ACTIN* forward primer: 5′-CATGTACGTTGCTATCCAGGC-3′
      iv. *ACTIN* reverse primer: 5′-CTCCTTAATGTCACGCACGAT-3′
      v. *36B4* forward primer: 5′-GTGTTCGACAATGGCAGCAT-3′
      vi. *36B4* reverse primer: 5′-GACACCCTCCAGGAAGCGA-3′
   c. *36B4* primer sequences reported in *Fullwood et al. (2009)*.
   d. Do not pre-treat with uracil-N-glycosylase.
   e. All reactions should be optimized and run in technical triplicate.
8. Using *ACTIN* as an internal standard, calculate the fold change in *PTEN1P* expression relative to pCMV expressing cells using the comparative Ct method.
   a. Additionally perform normalization using *36B4* as an internal standard (additional control).
9. Repeat independently two additional times.

## Deliverables

- Data to be collected:
  - Purity ($A_{260/280}$ ratio) and concentration of isolated total RNA from cells.
  - Raw data for all qPCR reactions.
  - Quantification of *PTENP1* mRNA levels relative to *ACTIN* or *36B4*.
  - Quantification of fold change *PTENP1* mRNA levels relative to pCMV transfected cells. (Compare to Figure 4A, left panel).

## Confirmatory analysis plan

This replication attempt will perform the following statistical analysis listed below.

- Statistical Analysis:
  - Note: at the time of analysis, we will perform the Shapiro–Wilk test and generate a quantile–quantile plot to assess the normality of the data. We will also perform Levene's test to assess homoscedasticity. If the data appear skewed we will perform the appropriate transformation in order to proceed with the proposed statistical analysis. If this is not possible we will perform the equivalent non-parametric test.
  - Unpaired two-tailed *t*-test of *PTENP1* mRNA levels of pCMV transfected cells compared to pCMV/*PTEN* 3′ UTR transfected cells.

- ■ Meta-analysis of effect sizes:
  - ○ Compute the effect sizes of each comparison, compare them against the effect size in the original paper and use a random effects meta-analytic approach to combine the original and replication effects, which will be presented as a forest plot.
- ■ Additional exploratory analysis:
  - ○ The same analysis described above will be performed with *36B4* normalized values, which serves as an independent normalization control not included in the original analysis.

## Known differences from the original study

The *PTENP1* mRNA levels will be normalized with an independent control (*36B4*) in addition to *ACTIN*. All known differences are listed in the materials and reagents section above with the originally used item listed in the comments section. All differences have the same capabilities as the original and are not expected to alter the experimental design.

## Provisions for quality control

The cell line used in this experiment will undergo STR profiling to confirm their identity and will be sent for mycoplasma testing to ensure there is no contamination. The sample purity ($A_{260/280}$ ratio) of the isolated RNA from each sample will be reported. The *PTENP1* mRNA levels will be normalized with an independent control (*36B4*). All the raw data, including the analysis files, will be uploaded to the project page on the OSF (https://osf.io/yyqas) and made publically available.

## Protocol 6: Cell growth assay following PTEN 3′ UTR transfection

This experiment tests the effect of expressing the 3′ UTR of PTENP1 on cell growth. It is a replication of the right panel of Figure 4A.

### Sampling

- ■ Experiment to be repeated a total of three times for a minimum power of 98%.
  - ○ See 'Power calculations' section for details.
- ■ Experiment has 3 conditions:
  - ○ Cohort 1: pCMV transfected DU145 cells.
  - ○ Cohort 2: pCMV/*PTEN* 3′ UTR transfected DU145 cells.
  - ○ Cohort 3: Uninfected DU145 cells (additional negative control).
- ■ Each cohort is harvested on the following days performed in technical triplicate:
  - ○ Day 0 (after O/N incubation).
  - ○ Day 1.
  - ○ Day 2.
  - ○ Day 3.
  - ○ Day 4.
  - ○ Day 5.

### Materials and reagents

| Reagent | Type | Manufacturer | Catalog # | Comments |
|---|---|---|---|---|
| DU145 cells | Cell line | ATCC | HTB-81 | – |
| RPMI 1640 medium | Cell culture | Sigma–Aldrich | R8758 | Replaces Invitrogen brand used in original study |
| Fetal bovine serum (FBS) | Cell culture | Sigma–Aldrich | F2442 | Replaces Invitrogen brand used in original study |
| L-glutamine | Cell culture | Sigma–Aldrich | G7513 | Original brand not specified |
| 100× Penicillin/streptomycin | Cell culture | Sigma–Aldrich | P4333 | Original brand not specified |
| 0.05% trypsin/0.48 mM EDTA | Cell culture | Sigma–Aldrich | T3924 | Original brand not specified |
| Phosphate buffered saline (PBS), without MgCl$_2$ and CaCl$_2$ | Cell culture | Sigma–Aldrich | D8537 | Original brand not specified |
| 60 mm tissue culture dishes | Labware | Corning | 430166 | Original brand not specified |
| pCMV (empty vector) | DNA construct | Original lab | n/a | From original lab |

*Continued on next page*

*Continued*

| Reagent | Type | Manufacturer | Catalog # | Comments |
|---|---|---|---|---|
| pCMV/*PTEN* 3′ UTR | DNA construct | Original lab | n/a | From original lab |
| Effectene | Cell culture | Qiagen | 301425 | Original brand not specified |
| 12 well tissue culture dishes | Labware | Corning | 3513 | Original brand not specified |
| Crystal violet | Dye | Sigma–Aldrich | C0775 | Original brand not specified |
| Formalin | Chemical | Specific brand information will be left up to the discretion of the replicating lab and recorded later | | |
| Acetic acid | Chemical | | | |
| Methanol | Chemical | | | |
| Spectrophotometer capable of reading at 590 nm (or 595 nm) | Instrument | Alamolabs | | Original brand not specified |

## Procedure

### Note

- All cells will be sent for mycoplasma testing and STR profiling.
- DU145 cells grown in complete RPMI 1640: RPMI 1640 supplemented with 2 mM glutamine, 10% FBS, 100 U/ml penicillin and 100 µg/ml streptomycin at 37°C and 6% $CO_2$.

1. Seed $3.5 \times 10^5$ DU145 cells per dish in 6 cm dishes. Grow overnight.
2. Transfect with pCMV or pCMV/*PTEN* 3′ UTR plasmids using Effectene according to manufacturer's instructions and recommended DNA and reagent amounts.
   a. Plasmids prepped in Protocol 7.
3. 6 hr after transfection, resuspend $2 \times 10^5$ pCMV, pCMV/*PTEN* 3′ UTR, and untransfected cells in 50 ml fresh media. Seed three wells of six sets of 12-well plates with 2 ml of each cell line. Each set of 12 well plates should have three wells containing untransfected cells, three wells containing pCMV-transfected cells, and three wells containing pCMV/PTEN 3′ UTR-transfected cells. Incubate overnight.
4. Fix one plate every 24 hr starting after overnight incubation (the first plate fixed will be called day 0).
   a. Wash wells once in PBS.
   b. Fix wells with 10% formalin for 10 min at room temperature.
   c. Store plates in PBS at 4°C.
   d. All wells should be fixed by day 6.
5. Stain cells with 0.1% crystal violet, 20% methanol for 15 min. Wash cells.
6. Lyse all wells with 10% acetic acid for 10 min.
7. Read optical density at 590 nm.
   a. Reading can be done at 595 nm if 590 nm is not available.
8. Repeat independently two additional times.

### Deliverables

- Data to be collected:
  - Raw data of absorbance from plate reader.
  - Relative absorbance for each cohort over time. (Compare to Figure 4A, right panel).

### Confirmatory analysis plan
This replication attempt will perform the following statistical analysis listed below.

- Statistical Analysis:
  - Note: at the time of analysis, we will perform the Shapiro–Wilk test and generate a quantile–quantile plot to assess the normality of the data. We will also perform Levene's test to assess homoscedasticity. If the data appear skewed we will perform the appropriate transformation in order to proceed with the proposed statistical analysis. If this is not possible we will perform the equivalent non-parametric test.
  - Unpaired two-tailed *t*-test of Day 5 absorbance of pCMV transfected cells compared to pCMV/*PTEN* 3′ UTR transfected cells.

○ Unpaired two-tailed *t*-test of AUC measurements (determined from day 0, 1, 2, 3, 4, and 5 for each replicate) of pCMV transfected cells compared to pCMV/*PTEN* 3′ UTR transfected cells.
- Meta-analysis of effect sizes:
  ○ Compute the effect sizes of each comparison, compare them against the effect size in the original paper and use a random effects meta-analytic approach to combine the original and replication effects, which will be presented as a forest plot.

## Known differences from the original study

All known differences are listed in the materials and reagents section above with the originally used item listed in the comments section. All differences have the same capabilities as the original and are not expected to alter the experimental design.

## Provisions for quality control

The cell line used in this experiment will undergo STR profiling to confirm their identity and will be sent for mycoplasma testing to ensure there is no contamination. All the raw data, including the analysis files, will be uploaded to the project page on the OSF (https://osf.io/yyqas) and made publically available.

# Power calculations

For additional details on power calculations, please see analysis scripts and associated files on the OSF:

   https://osf.io/cd2yq/

## Protocol 1

Summary of original data estimated from graph reported in Figure 1D:

| siRNA | mRNA | Mean | Stdev | N |
|---|---|---|---|---|
| siLUC | *PTEN* | 1.00 | 0.239 | 3 |
| | *PTENP1* | 1.00 | 0.386 | 3 |
| 19b | *PTEN* | 0.286 | 0.085 | 3 |
| | *PTENP1* | 0.234 | 0.065 | 3 |
| 20a | *PTEN* | 0.458 | 0.167 | 3 |
| | *PTENP1* | 0.250 | 0.080 | 3 |

## Test family

- 2 tailed *t* test, Wilcoxon–Mann-Whitney test, Bonferroni's correction: alpha error = 0.0125.

   'Power calculations' performed with G*Power software, version 3.1.7 (*Faul et al., 2007*).

| Group 1 | Group 2 | Effect size *d* | A priori power | Group 1 sample size | Group 2 sample size |
|---|---|---|---|---|---|
| siLuc *PTEN* mRNA | 19b *PTEN* mRNA | 3.98555 | 91.9%* | 4* | 4* |
| siLuc *PTEN* mRNA | 20a *PTEN* mRNA | 2.62999 | 88.3% | 6 | 6 |
| siLuc *PTENP1* mRNA | 19b *PTENP1* mRNA | 2.76532 | 80.7%† | 5† | 5† |
| siLuc *PTENP1* mRNA | 20a *PTENP1* mRNA | 2.68908 | 89.8% | 6 | 6 |

*6 samples per group will be used based on the siLuc to 20a *PTEN* comparison making the power 99.9%.
†6 samples per group will be used based on the siLuc to 20a *PTEN* comparison making the power 91.6%.

## Test family

- Due to the large variance, these parametric tests are only used for comparison purposes. The sample size is based on the non-parametric tests listed above.
- Two-way ANOVA: Fixed effects, special, main effects and interactions: alpha error = 0.05.

○ Due to a lack of raw original data, we are unable to perform power calculations using a MANOVA. We are using a two-way ANOVA to estimate sample size.

'Power calculations' performed with G*Power software, version 3.1.7 (*Faul et al., 2007*). ANOVA F test statistic and partial $\eta^2$ performed with R software, version 3.1.2 (*R Development Core Team, 2014*).

| Groups | F test statistic | Partial $\eta^2$ | Effect size $f$ | A priori power | Total sample size |
|---|---|---|---|---|---|
| siLUC, 19b, 20a (*PTEN* and *PTENP1* mRNA for all) | F(2,12) = 23.1978 (main effect: siRNA) | 0.79451 | 1.96629 | 96.3%* | 9* (6 groups) |

*36 total samples (6 per group) will be used based on the planned comparisons making the power 99.9%.

## Test family

- Due to the large variance, these parametric tests are only used for comparison purposes. The sample size is based on the non-parametric tests listed above.
- 2 tailed $t$ test, difference between two independent means, Bonferroni's correction: alpha error = 0.0125.

'Power calculations' performed with G*Power software, version 3.1.7 (*Faul et al., 2007*).

| Group 1 | Group 2 | Effect size $d$ | A priori power | Group 1 sample size | Group 2 sample size |
|---|---|---|---|---|---|
| siLuc *PTEN* mRNA | 19b *PTEN* mRNA | 3.98555 | 94.2%* | 4* | 4* |
| siLuc *PTEN* mRNA | 20a *PTEN* mRNA | 2.62999 | 90.6% | 6 | 6 |
| siLuc *PTENP1* mRNA | 19b *PTENP1* mRNA | 2.76532 | 84.0%† | 5† | 5† |
| siLuc *PTENP1* mRNA | 20a *PTENP1* mRNA | 2.68908 | 81.6%‡ | 5‡ | 5‡ |

*6 samples per group will be used based on the siLuc to 20a *PTEN* comparison making the power 99.9%.
†6 samples per group will be used based on the siLuc to 20a *PTEN* comparison making the power 93.4%.
‡6 samples per group will be used based on the siLuc to 20a *PTEN* comparison making the power 91.9%.

## Protocol 2

Summary of original data estimated from graph reported in Figure 2F:

| siRNA | Day | Mean | Stdev | N |
|---|---|---|---|---|
| siLuc | 0 | 1.000 | 0 | 3 |
| | 1 | 0.955 | 0.108 | 3 |
| | 2 | 0.928 | 0.108 | 3 |
| | 3 | 1.252 | 0.108 | 3 |
| | 4 | 1.315 | 0.108 | 3 |
| | 5 | 1.604 | 0.108 | 3 |
| siPTEN | 0 | 1.000 | 0 | 3 |
| | 1 | 1.198 | 0.108 | 3 |
| | 2 | 1.306 | 0.108 | 3 |
| | 3 | 2.045 | 0.108 | 3 |
| | 4 | 2.414 | 0.108 | 3 |
| | 5 | 4.153 | 0.234 | 3 |

*Continued on next page*

*Continued*

| siRNA | Day | Mean | Stdev | N |
|---|---|---|---|---|
| siPTENP1 | 0 | 1.000 | 0 | 3 |
| | 1 | 1.162 | 0.108 | 3 |
| | 2 | 1.099 | 0.108 | 3 |
| | 3 | 1.613 | 0.108 | 3 |
| | 4 | 1.775 | 0.108 | 3 |
| | 5 | 2.613 | 0.108 | 3 |
| siPTEN/PTENP1 | 0 | 1.000 | 0 | 3 |
| | 1 | 1.198 | 0.108 | 3 |
| | 2 | 1.387 | 0.108 | 3 |
| | 3 | 2.396 | 0.108 | 3 |
| | 4 | 3.099 | 0.108 | 3 |
| | 5 | 5.414 | 0.171 | 3 |

AUC calculations from estimated values.
Calculations performed with R software 3.1.2 (*R Development Core Team, 2014*).

| siRNA | Days | Mean | Stdev | N |
|---|---|---|---|---|
| siLuc | 0, 1, 2, 3, 4, 5 | 5.752 | 0.486 | 3 |
| siPTEN | 0, 1, 2, 3, 4, 5 | 9.541 | 0.550 | 3 |
| siPTENP1 | 0, 1, 2, 3, 4, 5 | 7.455 | 0.486 | 3 |
| siPTEN/PTENP1 | 0, 1, 2, 3, 4, 5 | 11.288 | 0.518 | 3 |

## Test family

■ Two-way ANOVA: Fixed effects, special, main effects and interactions: alpha error = 0.05.

'Power calculations' performed with G*Power software, version 3.1.7 (*Faul et al., 2007*).
ANOVA F test statistic and partial $\eta^2$ performed with R software, version 3.1.2 (*R Development Core Team, 2014*).

### Day 5 values

| Groups | F test statistic | Partial $\eta^2$ | Effect size $f$ | A priori power | Total sample size |
|---|---|---|---|---|---|
| siLUC, siPTEN, siPTENP1, siPTEN/PTENP1 | F(1,8) = 798.9603 (main effect: siPTEN) | 0.99009 | 9.99337 | 92.0%* | 5* (4 groups) |
| | F(1,8) = 143.7867 (main effect: siPTENP1) | 0.94729 | 4.23948 | 99.5%* | 6* (4 groups) |

*16 total samples (4 per group) will be used based on the planned comparisons making the power 99.9%.

### AUC values

| Groups | F test statistic | Partial $\eta^2$ | Effect size $f$ | A priori power | Total sample size |
|---|---|---|---|---|---|
| siLUC, siPTEN, siPTENP1, siPTEN/PTENP1 | F(1,8) = 166.9731 (main effect: siPTEN) | 0.95428 | 4.56857 | 99.8%* | 6* (4 groups) |
| | F(1,8) = 34.2219 (main effect: siPTENP1) | 0.81053 | 2.06827 | 93.9%* | 7* (4 groups) |

*16 total samples (4 per group) will be used based on the planned comparisons making the power 99.9%.

## Test family

- 2 tailed *t* test, difference between two independent means, Bonferroni's correction: alpha error = 0.01.

  'Power calculations' performed with G*Power software, version 3.1.7 (*Faul et al., 2007*).

## Day 5 values

| Group 1 | Group 2 | Effect size *d* | A priori power | Group 1 sample size | Group 2 sample size |
|---------|---------|-----------------|----------------|---------------------|---------------------|
| siLuc | siPTEN | 15.54994 | 91.1%* | 2* | 2* |
| siLuc | siPTENP1 | 6.15404 | 91.1%* | 3* | 3* |
| siLuc | siPTEN/PTENP1 | 23.24249 | 99.9%* | 2* | 2* |
| siPTEN/PTENP1 | siPTEN | 7.69255 | 99.4%* | 3* | 3* |
| siPTEN/PTENP1 | siPTENP1 | 17.08845 | 94.6%* | 2* | 2* |

*5 samples per group will be used based on the AUC calculation planned comparisons making the power 99.9%.

## AUC values

| Group 1 | Group 2 | Effect size *d* | A priori power | Group 1 sample size | Group 2 sample size |
|---------|---------|-----------------|----------------|---------------------|---------------------|
| siLuc | siPTEN | 7.41636 | 99.1%* | 3* | 3* |
| siLuc | siPTENP1 | 3.33339 | 93.8% | 5 | 5* |
| siLuc | siPTEN/PTENP1 | 10.83794 | 99.9%* | 3* | 3* |
| siPTEN/PTENP1 | siPTEN | 3.42158 | 81.3%† | 4† | 4† |
| siPTEN/PTENP1 | siPTENP1 | 7.50454 | 99.3%* | 3* | 3* |

*5 samples per group will be used based on the siLuc to siPTENP1 comparison making the power 99.9%.
†5 samples per group will be used based on the siLuc to siPTENP1 comparison making the power 95.0%.

## Protocol 3

Summary of original data estimated from graph reported in Figure 2G:

| siRNA | mRNA | Mean | Stdev | N |
|-------|------|------|-------|---|
| siLUC | *PTEN* | 1.000 | 0.249 | 3 |
| | *PTENP1* | 1.000 | 0.155 | 3 |
| siPTEN | *PTEN* | 0.116 | 0.065 | 3 |
| | *PTENP1* | 0.543 | 0.099 | 3 |
| siPTENP1 | *PTEN* | 0.381 | 0.086 | 3 |
| | *PTENP1* | 0.269 | 0.094 | 3 |
| siPTEN/PTENP1 | *PTEN* | 0.193 | 0.067 | 3 |
| | *PTENP1* | 0.482 | 0.161 | 3 |

## Test family

- 2 tailed *t* test, Wilcoxon–Mann-Whitney test, Bonferroni's correction: alpha error = 0.008333.

  'Power calculations' performed with G*Power software, version 3.1.7 (*Faul et al., 2007*).

## Test family

- Due to the large variance, these parametric tests are only used for comparison purposes. The sample size is based on the non-parametric tests listed above.
- Two-way ANOVA: Fixed effects, special, main effects and interactions: alpha error = 0.05.

| Group 1 | Group 2 | Effect size d | A priori power | Group 1 sample size | Group 2 sample size |
|---|---|---|---|---|---|
| siLuc *PTEN* mRNA | siPTEN *PTEN* mRNA | 4.85884 | 96.8%* | 4* | 4* |
| siLuc *PTENP1* mRNA | siPTEN *PTENP1* mRNA | 3.52537 | 93.1% | 5 | 5 |
| siLuc *PTEN* mRNA | siPTENP1 *PTEN* mRNA | 3.32267 | 89.7% | 5 | 5 |
| siLuc *PTENP1* mRNA | siPTENP1 *PTENP1* mRNA | 5.70745 | 83.9%† | 3† | 3† |
| siLuc *PTEN* mRNA | siPTEN/PTENP1 *PTEN* mRNA | 4.42658 | 93.2%‡ | 4‡ | 4‡ |
| siLuc *PTENP1* mRNA | siPTEN/PTENP1 *PTENP1* mRNA | 3.27585 | 88.7% | 5 | 5 |

*5 samples per group will be used based on the siLuc to siPTENP1 *PTEN* comparison making the power 99.8%.
†5 samples per group will be used based on the siLuc to siPTENP1 *PTEN* comparison making the power 99.9%.
‡5 samples per group will be used based on the siLuc to siPTENP1 *PTEN* comparison making the power 99.3%.

○ Due to a lack of raw original data, we are unable to perform power calculations using a MANOVA. We are using a two-way ANOVA to estimate sample size.

| Groups | F test statistic | Partial $\eta^2$ | Effect size f | A priori power | Total sample size |
|---|---|---|---|---|---|
| siLUC, siPTEN, siPTENP1, siPTEN/PTENP1 (*PTEN* and *PTENP1* mRNA for all) | F(3,16) = 36.6570 (main effect: siRNA) | 0.87299 | 2.62168 | 94.9%* | 11* (8 groups) |

*40 total samples (5 per group) will be used based on the planned comparisons making the power 99.9%.

'Power calculations' performed with G*Power software, version 3.1.7 (*Faul et al., 2007*). ANOVA F test statistic and partial $\eta^2$ performed with R software, version 3.1.2 (*R Development Core Team, 2014*).

## Test family

■ Due to the large variance, these parametric tests are only used for comparison purposes. The sample size is based on the non-parametric tests listed above.

| Group 1 | Group 2 | Effect size d | A priori power | Group 1 sample size | Group 2 sample size |
|---|---|---|---|---|---|
| siLuc *PTEN* mRNA | siPTEN *PTEN* mRNA | 4.85884 | 98.1%* | 4* | 4* |
| siLuc *PTENP1* mRNA | siPTEN *PTENP1* mRNA | 3.52537 | 80.8%† | 4† | 4† |
| siLuc *PTEN* mRNA | siPTENP1 *PTEN* mRNA | 3.32267 | 92.2% | 5 | 5 |
| siLuc *PTENP1* mRNA | siPTENP1 *PTENP1* mRNA | 5.70745 | 89.1%‡ | 3‡ | 3‡ |
| siLuc *PTEN* mRNA | siPTEN/PTENP1 *PTEN* mRNA | 4.42658 | 95.4%§ | 4§ | 4§ |
| siLuc *PTENP1* mRNA | siPTEN/PTENP1 *PTENP1* mRNA | 3.27585 | 91.4% | 5 | 5 |

*5 samples per group will be used based on the siLuc to siPTENP1 *PTEN* comparison making the power 99.9%.
†5 samples per group will be used based on the siLuc to siPTENP1 *PTEN* comparison making the power 95.1%.
‡5 samples per group will be used based on the siLuc to siPTENP1 *PTEN* comparison making the power 99.9%.
§5 samples per group will be used based on the siLuc to siPTENP1 *PTEN* comparison making the power 99.6%.

■ 2 tailed *t* test, difference between two independent means, Bonferroni's correction: alpha error = 0.008333.

'Power calculations' performed with G*Power software, version 3.1.7 (*Faul et al., 2007*).

| siRNA | Relative PTEN signal |
|---|---|
| siLuc | 1.00 |
| siPTEN | 0.50 |
| siPTENP1 | 0.60 |
| siPTEN/PTENP1 | 0.10 |

## Protocol 4

Summary of original data reported in Figure 2H:

The original data do not indicate the error associated with multiple biological replicates. To identify a suitable sample size, power calculations were performed using different levels of relative variance.

### Test family

- Two-way ANOVA: Fixed effects, special, main effects and interactions: alpha error = 0.05.

'Power calculations' performed with G*Power software, version 3.1.7 (*Faul et al., 2007*).
ANOVA F test statistic and partial $\eta^2$ performed with R software, version 3.1.2 (*R Development Core Team, 2014*).

2% variance:

| Groups | F test statistic | Partial $\eta^2$ | Effect size $f$ | A priori power | Total sample size |
|---|---|---|---|---|---|
| siLUC, siPTEN, siPTENP1, siPTEN/PTENP1 | $F(1,8) = 4629.6$ (main effect: siPTEN) | 0.99828 | 24.0564 | 99.9% | 8 (4 groups) |
| | $F(1,8) = 2963.0$ (main effect: siPTENP1) | 0.99731 | 19.24404 | 99.9% | 8 (4 groups) |

15% variance:

| Groups | F test statistic | Partial $\eta^2$ | Effect size $f$ | A priori power | Total sample size |
|---|---|---|---|---|---|
| siLUC, siPTEN, siPTENP1, siPTEN/PTENP1 | $F(1,8) = 82.3050$ (main effect: siPTEN) | 0.91141 | 3.20750 | 99.9% | 8 (4 groups) |
| | $F(1,8) = 52.6750$ (main effect: siPTENP1) | 0.86815 | 2.56600 | 99.9% | 8 (4 groups) |

28% variance:

| Groups | F test statistic | Partial $\eta^2$ | Effect size $f$ | A priori power | Total sample size |
|---|---|---|---|---|---|
| siLUC, siPTEN, siPTENP1, siPTEN/PTENP1 | $F(1,8) = 23.6210$ (main effect: siPTEN) | 0.74700 | 1.71830 | 94.5% | 8 (4 groups) |
| | $F(1,8) = 15.1170$ (main effect: siPTENP1) | 0.65394 | 1.37464 | 82.4% | 8 (4 groups) |

40% variance:

| Groups | F test statistic | Partial $\eta^2$ | Effect size $f$ | A priori power | Total sample size |
|---|---|---|---|---|---|
| siLUC, siPTEN, siPTENP1, siPTEN/PTENP1 | $F(1,8) = 11.5741$ (main effect: siPTEN) | 0.59130 | 1.20281 | 95.3% | 12 (4 groups) |
| | $F(1,8) = 7.4074$ (main effect: siPTENP1) | 0.48077 | 0.96225 | 83.0% | 12 (4 groups) |

### Test family

- 2 tailed *t* test, difference between two independent means, Bonferroni's correction: alpha error = 0.01.

'Power calculations' performed with G*Power software, version 3.1.7 (*Faul et al., 2007*).

2% variance:

| Group 1 | Group 2 | Effect size $d$ | A priori power | Group 1 sample size | Group 2 sample size |
|---|---|---|---|---|---|
| siLuc | siPTEN | 39.28399 | 99.9% | 2 | 2 |
| siLuc | siPTENP1 | 31.42719 | 99.9% | 2 | 2 |
| siLuc | siPTEN/PTENP1 | 70.71118 | 99.9% | 2 | 2 |
| siPTEN/PTENP1 | siPTEN | 31.42719 | 99.9% | 2 | 2 |
| siPTEN/PTENP1 | siPTENP1 | 39.28399 | 99.9% | 2 | 2 |

15% variance:

| Group 1 | Group 2 | Effect size $d$ | A priori power | Group 1 sample size | Group 2 sample size |
|---|---|---|---|---|---|
| siLuc | siPTEN | 5.23782 | 86.6% | 3 | 3 |

*Continued on next page*

*Continued*

| Group 1 | Group 2 | Effect size *d* | A priori power | Group 1 sample size | Group 2 sample size |
| --- | --- | --- | --- | --- | --- |
| siLuc | siPTENP1 | 4.19026 | 86.6% | 4 | 4 |
| siLuc | siPTEN/PTENP1 | 9.42808 | 99.9% | 3 | 3 |
| siPTEN/PTENP1 | siPTEN | 4.19026 | 94.6% | 4 | 4 |
| siPTEN/PTENP1 | siPTENP1 | 5.23782 | 86.6% | 3 | 3 |

28% variance:

| Group 1 | Group 2 | Effect size *d* | A priori power | Group 1 sample size | Group 2 sample size |
| --- | --- | --- | --- | --- | --- |
| siLuc | siPTEN | 2.80603 | 82.0% | 5 | 5 |
| siLuc | siPTENP1 | 2.24482 | 84.8% | 7 | 7 |
| siLuc | siPTEN/PTENP1 | 5.05085 | 84.0% | 3 | 3 |
| siPTEN/PTENP1 | siPTEN | 2.24482 | 84.8% | 7 | 7 |
| siPTEN/PTENP1 | siPTENP1 | 2.80603 | 82.0% | 5 | 5 |

40% variance:

| Group 1 | Group 2 | Effect size *d* | A priori power | Group 1 sample size | Group 2 sample size |
| --- | --- | --- | --- | --- | --- |
| siLuc | siPTEN | 1.96419 | 86.9% | 8 | 8 |
| siLuc | siPTENP1 | 1.57135 | 84.7% | 12 | 12 |
| siLuc | siPTEN/PTENP1 | 3.53554 | 91.0% | 4 | 4 |
| siPTEN/PTENP1 | siPTEN | 1.57135 | 83.6% | 12 | 12 |
| siPTEN/PTENP1 | siPTENP1 | 1.96419 | 81.0% | 8 | 8 |

In order to produce quantitative replication data, we will run the experiment five times. Each time we will quantify band intensity. We will determine the standard deviation of band intensity across the biological replicates and combine this with the reported value from the original study to simulate the original effect size. We will use this simulated effect size to determine the number of replicates necessary to reach a power of at least 80%. We will then perform additional replicates, if required, to ensure that the experiment has more than 80% power to detect the original effect.

## Protocol 5

Summary of original data estimated from graph reported in Figure 4A, left panel:

| Plasmid | Mean | Stdev | N |
| --- | --- | --- | --- |
| pCMV | 1.000 | 0.541 | 3 |
| pCMV/*PTEN* 3′ UTR | 3.880 | 0.707 | 3 |

### Test family

- 2 tailed *t* test, difference between two independent means, alpha error = 0.05.

  'Power calculations' performed with G*Power software, version 3.1.7 (*Faul et al., 2007*).

| Group 1 | Group 2 | Effect size *d* | A priori power | Group 1 sample size | Group 2 sample size |
| --- | --- | --- | --- | --- | --- |
| pCMV | pCMV/*PTEN* 3′ UTR | 4.57446 | 98.2% | 3 | 3 |

## Protocol 6

Summary of original data estimated from graph reported in Figure 4A, right panel:

| Plasmid | Day | Mean | Stdev | N |
|---|---|---|---|---|
| pCMV | 0 | 1.000 | 0 | 3 |
| | 1 | 1.000 | 0.119 | 3 |
| | 2 | 1.238 | 0.119 | 3 |
| | 3 | 1.917 | 0.119 | 3 |
| | 4 | 5.726 | 0.119 | 3 |
| | 5 | 7.167 | 0.119 | 3 |
| pCMV/PTEN 3′ UTR | 0 | 1.000 | 0 | 3 |
| | 1 | 1.000 | 0.119 | 3 |
| | 2 | 1.143 | 0.119 | 3 |
| | 3 | 1.917 | 0.119 | 3 |
| | 4 | 4.155 | 0.214 | 3 |
| | 5 | 5.238 | 0.119 | 3 |

AUC calculations from estimated values.
Calculations performed with R software 3.1.2 (*R Development Core Team, 2014*).

| Plasmid | Days | Mean | Stdev | N |
|---|---|---|---|---|
| pCMV | 0, 1, 2, 3, 4, 5 | 13.964 | 0.536 | 3 |
| pCMV/PTEN 3′ UTR | 0, 1, 2, 3, 4, 5 | 11.333 | 0.631 | 3 |

## Test family

- 2 tailed *t* test, difference between two independent means, alpha error = 0.05.

'Power calculations' performed with G*Power software, version 3.1.7 (*Faul et al., 2007*).

## Day 5 values

| Group 1 | Group 2 | Effect size *d* | A priori power | Group 1 sample size | Group 2 sample size |
|---|---|---|---|---|---|
| pCMV | pCMV/PTEN 3′ UTR | 16.20000 | 99.9% | 2* | 2* |

*3 samples per group will be used based on the AUC calculation.

## AUC values

| Group 1 | Group 2 | Effect size *d* | A priori power | Group 1 sample size | Group 2 sample size |
|---|---|---|---|---|---|
| pCMV | pCMV/PTEN 3′ UTR | 4.49526 | 97.9% | 3 | 3 |

## Acknowledgements

The Reproducibility Project: Cancer Biology core team would like to thank the original authors, in particular Laura Poliseno, Leonardo Salmena, and Pier Paolo Pandolfi, for generously sharing critical information as well as reagents to ensure the fidelity and quality of this replication attempt. We thank Courtney Soderberg at the Center for Open Science for assistance with statistical analyses. We would also like to thank the following companies for generously donating reagents to the Reproducibility Project: Cancer Biology; American Type Culture Collection (ATCC), Applied Biological Materials, BioLegend, Charles River Laboratories, Corning Incorporated, DDC Medical, EMD Millipore, Harlan Laboratories, LI-COR Biosciences, Mirus Bio, Novus Biologicals, Sigma–Aldrich, and System Biosciences (SBI). We thank Dale Cowley and Kumar Pandya, TransViragen, Inc., for technical suggestions related to experiments to be performed.

## Additional information

### Group author details

**Reproducibility Project: Cancer Biology**

Elizabeth Iorns: Science Exchange, Palo Alto, California; William Gunn: Mendeley, London, United Kingdom; Fraser Tan: Science Exchange, Palo Alto, California; Joelle Lomax: Science Exchange, Palo Alto, California; Nicole Perfito: Science Exchange, Palo Alto, California; Timothy Errington: Center for Open Science, Charlottesville, Virginia

### Competing interests

IK: Alamo Laboratories Inc. is a Science Exchange associated laboratory. JK: Biotechnology Research and Education Program, University of Maryland is a Science Exchange associated laboratory. RP:CB: EI, FT, JL, and NP are employed and holds shares in Science Exchange Inc. The other authors declare that no competing interests exist.

### Funding

| Funder | Author |
| --- | --- |
| Laura and John Arnold foundation | Reproducibility Project: Cancer Biology |

The Reproducibility Project: Cancer Biology is funded by the Laura and John Arnold Foundation, provided to the Center for Open Science in collaboration with Science Exchange. The funder had no role in study design or the decision to submit the work for publication.

### Author contributions

IK, JK, KO, EG, Drafting or revising the article; RP:CB, Conception and design, Drafting or revising the article

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
