## [Decision Letter]

Thank you for submitting your work entitled “Registered report: A coding-independent function of gene and pseudogene mRNAs regulates tumour biology” for peer review at *eLife*. Your submission has been favorably evaluated by Charles Sawyers (Senior Editor), and three reviewers, one of whom is a member of our Board of Reviewing Editors.

The reviewers have discussed their comments with one another, and the Reviewing Editor has drafted this decision to help you prepare a revised submission.

This Registered Report describes a proposed replication study that is focused on Poliseno et al. (2010, Nature 465, 1033-1038). The central conclusion of [26] is that expression of an endogenous miRNA competitor (e.g. a pseudogene) can de-repress miRNA function.

There are four important points that the authors should address:

1) The Introduction lays out the background, the results of [26], and consistent data reported in other studies. However, the Introduction should include other studies that lead to questions concerning the ceRNA hypothesis (e.g. PMID: 24793693 & PMID: 24905003 and references therein). The ceRNA hypothesis remains an active area for inquiry. The impression given by the Introduction is that the overwhelming balance of evidence favors the ceRNA hypothesis – this is not correct.

2) It is well established that over-expression of miRNA can repress target mRNA (Figure 1D) and that over-expression of a miRNA sponge can de-repress a miRNA target (Figure 2A-D and Figure 4A). While repetition of these figures panels is useful, none of these experiments tackle the central conclusions of [26] – that expression of an endogenous miRNA competitor (e.g. a pseudogene) can de-repress miRNA function. This is because of the reliance on over-expression strategies in Figure 1D, Figure 2A-D, and Figure 4A. Studies of the endogenous genes are required to test reproducibility of the central conclusion of the original study.

The major conclusion of [26] is that decreased expression of *PTEN* or *PTENP1* mRNA should cause mutual changes in miRNA-mediated repression. The only experimental test on endogenous *PTEN* and *PTENP1* mRNA (rather than over-expression of cDNA or miRNA) is represented by Figure 2F-H. It is therefore essential that Figure 2F-H are the focus of the replication study because these are the only experiments that address the central conclusion of [26].

3) The normalization used for the mRNA measurements by [26] and this Registered Report is *ACTIN*. This is a problem because Actin mRNA is dynamically regulated in cells and is a target of cell signaling pathway. Perhaps the *ACTIN* measurements can be retained in the replication study, but we would strongly recommend that an independent normalization control is also employed – for example *36B4* or another control gene.

4) The authors adopt two statistical procedures, ANOVA and 2 tailed *t*-test, in the analysis. However, neither of them is used correctly. Take Protocol 1 as an example.

A) In ANOVA, for testing the main effect of siRNA, there are only 3 different levels: siLUC, *19b* and *20a*, but the numerator degree of freedom of the F-test is stated to be 4 by the authors. I also couldn't follow how the authors got the denominator d.f. to be 20 given there are only 18 observations in total.

B) In 2 tailed *t*-test, the effective size calculated by the authors assumed that the two populations have the same variance, which is apparently not true for many pairs, according to the data. For example, in the comparison of siLuc *PTENP1* and *19b PTENP1*, the two sample standard deviations are 0.386 and 0.065, respectively. The former is almost 6-fold of the later. Moreover, given this small sample size, *t*-test is not recommended unless one is very confident that the two population distributions are normal.

5) Besides the paper that the authors intend to replicate, the oncosuppressive properties of *PTEN* 3'UTR have been shown also in Tay et al. (Cell, 2015), a paper which should be quoted. Analogously, the oncosuppressive properties of *PTENP1* have been shown in other papers besides [41], namely [6] and [15].

6) In more general terms, we suggest that the Introduction should include a comprehensive review of all the papers in which the ceRNA-based relationship between *PTEN* and *PTENP1* has been confirmed:

Poliseno, L., et al., Deletion of *PTENP1* pseudogene in human melanoma. J Invest Dermatol, 2011. 131(12): p. 2497-500.

Ioffe, Y.J., et al., Phosphatase and tensin homolog (*PTEN*) pseudogene expression in endometrial cancer: a conserved regulatory mechanism important in tumorigenesis? Gynecol Oncol, 2012. 124(2): p. 340-6.

Johnsson, P., et al., A pseudogene long-noncoding-RNA network regulates *PTEN* transcription and translation in human cells. Nat Struct Mol Biol, 2013. 20(4): p. 440-6.

Yu, G., et al., Pseudogene *PTENP1* functions as a competing endogenous RNA to suppress clear cell renal cell carcinoma progression. Mol Cancer Ther, 2014.

Chen, C.L., et al., Suppression of hepatocellular carcinoma by baculovirus-mediated expression of long non-coding RNA *PTENP1* and MicroRNA regulation. Biomaterials, 2015. 44: p. 71-81.

Guo, X., et al., Pseudogene *PTENP1* Suppresses Gastric Cancer Progression by Modulating *PTEN*. Anticancer Agents Med Chem, 2015.

---

## [Author Response]

*1) The Introduction lays out the background, the results of*
[26]*, and consistent data reported in other studies. However, the Introduction should include other studies that lead to questions concerning the ceRNA hypothesis (e.g. PMID: 24793693 & PMID: 24905003 and references therein). The ceRNA hypothesis remains an active area for inquiry. The impression given by the Introduction is that the overwhelming balance of evidence favors the ceRNA hypothesis – this is not correct*.

We have updated the Introduction to include these references and reflect a more balanced view of the ceRNA hypothesis.

*2) It is well established that over-expression of miRNA can repress target mRNA (Figure 1D) and that over-expression of a miRNA sponge can de-repress an miRNA target (Figure 2A-D and Figure 4A). While repetition of these figures panels is useful, none of these experiments tackle the central conclusions of*
[26]
*– that expression of an endogenous miRNA competitor (e.g. a pseudogene) can de-repress miRNA function. This is because of the reliance on over-expression strategies in Figure 1D, Figure 2A-D, and Figure 4A. Studies of the endogenous genes are required to test reproducibility of the central conclusion of the original study*.

*The major conclusion of*
[26]
*is that decreased expression of* PTEN *or* PTENP1 *mRNA should cause mutual changes in miRNA-mediated repression. The only experimental test on endogenous* PTEN *and* PTENP1 *mRNA (rather than over-expression of cDNA or miRNA) is represented by Figure 2F-H. It is therefore essential that Figure 2F-H are the focus of the replication study because these are the only experiments that address the central conclusion of*
[26].

Thank you for the suggestion. We have included Figures 2F-H in the revised manuscript. We are also removing Figures 2A, 2C, and 2D from the replication attempt.

*3) The normalization used for the mRNA measurements by*
[26]
*and the Registered Report is* ACTIN*. This is a problem because Actin mRNA is dynamically regulated in cells and is a target of cell signaling pathway. Perhaps the* ACTIN *measurements can be retained in the replication study, but I would strongly recommend that an independent normalization control is also employed – for example* 36B4 *or another control gene*.

Thank you for the suggestion. We have included *36B4* as an additional control gene. We will maintain the original analysis (using *ACTIN* measurements for normalization) and include an additional exploratory analysis from the original, using *36B4* to normalize the gene expression.

*4) The authors adopt two statistical procedures, ANOVA and 2 tailed* t*-test, in the analysis. However, neither of them is used correctly. Take Protocol 1 as an example*.

*A) In ANOVA, for testing the main effect of siRNA, there are only 3 different levels: siLUC,* 19b *and* 20a*, but the numerator degree of freedom of the F-test is stated to be 4 by the authors. I also couldn't follow how the authors got the denominator d.f. to be 20 given there are only 18 observations in total*.

Thank you for catching this error. We have gone through and checked all calculations. Regarding Protocol 1 in particular, the error with d.f. was an error upon transferring the information to the manuscript. The d.f. of the F-test is indeed (2,12) (3 levels with 18 total measurements). We have corrected any other location where this might have occurred in error.

*B) In 2 tailed* t*-test, the effective size calculated by the authors assumed that the two populations have the same variance, which is apparently not true for many pairs according to the data. For example, in the comparison of siLuc* PTENP1 *and* 19b PTENP1*, the two sample standard deviations are 0.386 and 0.065, respectively. The former is almost 6-fold of the later. Moreover, given this small sample size,* t*-test is not recommended unless one is very confident that the two population distributions are normal*.

Thank you for the comment. We have gone through and included non-parametric comparisons (Wilcoxon-Mann-Whitney) along with the student’s *t*-tests whenever the standard deviations are large. We have also included a statement in the confirmatory analysis plan where the normality and homoscedasticity of the data will be assessed prior to performing the proposed statistical analysis.

We have also included a link to the scripts, other files, and summary of calculations (https://osf.io/cd2yq/?view_only=c9c2497ac93c46b6a8ab786769fcff74).

*5) Besides the paper that the authors intend to replicate, the oncosuppressive properties of* PTEN *3'UTR have been shown also in Tay et al. (Cell, 2015), a paper which should be quoted. Analogously, the oncosuppressive properties of* PTENP1 *have been shown in other papers besides*
[41]*, namely*
[6]
*and*
[15].

We have updated the Introduction to reflect a more balanced view of the ceRNA hypothesis.

*6) In more general terms, we suggest that the Introduction should include a comprehensive review of all the papers in which the ceRNA-based relationship between* PTEN *and* PTENP1 *has been confirmed*:

*Poliseno, L., et al., Deletion of* PTENP1 *pseudogene in human melanoma. J Invest Dermatol, 2011. 131(12): p. 2497-500*.

*Ioffe, Y.J., et al., Phosphatase and tensin homolog (*PTEN*) pseudogene expression in endometrial cancer: a conserved regulatory mechanism important in tumorigenesis? Gynecol Oncol, 2012. 124(2): p. 340-6*.

*Johnsson, P., et al., A pseudogene long-noncoding-RNA network regulates* PTEN *transcription and translation in human cells. Nat Struct Mol Biol, 2013. 20(4): p. 440-6*.

*Yu, G., et al., Pseudogene* PTENP1 *functions as a competing endogenous RNA to suppress clear cell renal cell carcinoma progression. Mol Cancer Ther, 2014*.

*Chen, C.L., et al., Suppression of hepatocellular carcinoma by baculovirus-mediated expression of long non-coding RNA* PTENP1 *and MicroRNA regulation. Biomaterials, 2015. 44: p. 71-81*.

*Guo, X., et al., Pseudogene* PTENP1 *Suppresses Gastric Cancer Progression by Modulating* PTEN*. Anticancer Agents Med Chem, 2015*.

We have included a more balanced discussion of the literature of the ceRNA hypothesis. In the Introduction, we try to focus on the experiments being replicated and any direct replications of them, instead of a comprehensive review of all the literature on this subject, which includes many conceptual or related experiments, which would be more appropriate for a review.